# Randomized Clinical Trial of an Internet-Based Adolescent Depression Prevention Intervention in Primary Care: Internalizing Symptom Outcomes

**DOI:** 10.3390/ijerph17217736

**Published:** 2020-10-22

**Authors:** Tracy Gladstone, Katherine R. Buchholz, Marian Fitzgibbon, Linda Schiffer, Miae Lee, Benjamin W. Van Voorhees

**Affiliations:** 1The Robert S. and Grace W. Stone Primary Prevention Initiatives, Wellesley Centers for Women, Wellesley College, Wellesley, MA 02481, USA; katherine.buchholz@wellesley.edu; 2Department of General Pediatrics, College of Medicine, University of Illinois at Chicago, Chicago, IL 60612, USA; mlf@uic.edu (M.F.); mlee9@uic.edu (M.L.); bvanvoor@uic.edu (B.W.V.V.); 3Institute for Health Research and Policy, School of Public Health, University of Illinois at Chicago, Chicago, IL 60608, USA; lschiff@uic.edu; 4University of Illinois Cancer Center, University of Illinois at Chicago, Chicago, IL 60612, USA

**Keywords:** web-based interventions, internalizing symptoms, depressive symptoms, adolescents, prevention, primary care

## Abstract

Approximately 20% of people will experience a depressive episode by adulthood, making adolescence an important developmental target for prevention. CATCH-IT (Competent Adulthood Transition with Cognitive-behavioral, Humanistic, and Interpersonal Training), an online depression prevention intervention, has demonstrated efficacy in preventing depressive episodes among adolescents reporting elevated symptoms. Our study examines the effects of CATCH-IT compared to online health education (HE) on internalizing symptoms in adolescents at risk for depression. Participants, ages 13–18, were recruited across eight US health systems and were randomly assigned to CATCH-IT or HE. Assessments were completed at baseline, 2, 6, 12, 18, and 24 months. There were no significant differences between groups in change in depressive symptoms (b = −0.31 for CATCH-IT, b = −0.27 for HE, *p* = 0.80) or anxiety (b = −0.13 for CATCH-IT, b = −0.11 for HE, *p* = 0.79). Improvement in depressive symptoms was statistically significant (*p* < 0.05) for both groups (*p* = 0.004 for CATCH-IT, *p* = 0.009 for HE); improvement in anxiety was significant for CATCH-IT (*p* = 0.04) but not HE (*p* = 0.07). Parental depression and positive relationships with primary care physicians (PRPC) moderated the anxiety findings, and adolescents’ externalizing symptoms and PRPC moderated the depression findings. This study demonstrates the long-term positive effects of both online programs on depressive symptoms and suggests that CATCH-IT demonstrates cross-over effects for anxiety as well.

## 1. Introduction

Depression is a significant public health concern for adolescents. It is estimated that as many as 20% of people will experience a depressive episode before adulthood [1]. Depression is associated with significant impairments in day-to-day functioning, and it can affect developmental trajectories in adolescence and early adulthood [2,3,4]. Episodic in its course, a single depressive episode increases the risk of future episodes that are often more severe [2,3]. Depressive symptoms, including subthreshold symptoms, are associated with an increased risk of depressive episodes, physical health problems, comorbid mental health concerns, and suicide [4,5,6,7]. The personal and societal cost of depression can be eased through depression prevention interventions [8,9].

A recent Cochrane review revealed that depression prevention programs for adolescents are associated with a moderate reduction in depressive episodes (risk difference = −0.03) across one-year follow-up [10], and a meta-analysis of 19 randomized preventive trials revealed that, for adolescents with elevated symptoms of depression, preventive interventions are associated with benefits across a two-year follow-up [11]. Several trials of specific depression prevention approaches for adolescents at risk for depression also have been found to be effective for certain subsets of teens [12,13,14,15,16,17]. The majority of these interventions involve in-person groups that rely on training therapists and recruiting participants who can attend regular sessions; these approaches are limited by lack of access due to barriers related to seeking in-person interventions [14,15,16,17]. Therefore, it can be difficult to provide these interventions cost-effectively to the large number of adolescents who may benefit from them. Given the high rate of depression among adolescents and the long-term consequences of depression, public health interventions that can reach a broad population of adolescents are needed.

Similar to depression, anxiety is a significant public health problem among adolescents. In fact, anxiety, frequently grouped with depression as an internalizing disorder, is the most common mental health concern among adolescents [18,19]. The lifetime prevalence of anxiety disorders in adolescents is just over 30% [18]. Developmentally, anxiety has an earlier age of onset than depression [18,20,21]. Some studies have found that anxiety starting in childhood precedes depression that begins in adolescence [22,23], while other studies report that there is a bidirectional relationship between anxiety and depression in children and adolescents [24,25]. Comorbid presentations result in poorer overall prognosis [26]. With both disorders, onset during childhood or adolescence is associated with chronic symptoms or functional impairment in adulthood [27,28]. Anxiety often co-occurs with depression [29,30,31,32], and adolescents with depression have been found to be at greater risk for also having an anxiety disorder compared to adolescents without depression [32,33]. Epidemiological studies suggest that 30–75% of children or adolescents with depression have comorbid anxiety [33]. Therefore, prevention and treatment of depression in adolescents must consider anxiety.

Given the high rates of co-occurring depression and anxiety, research has focused on identifying transdiagnostic factors that may represent shared etiological mechanisms or latent factors that underlie both disorders [34]. These transdiagnostic factors include negative affectivity or emotionality [35], cognitive or information processing biases [36], emotional awareness [37], and emotional regulation skills [38]. Given that comorbidity is more common than just anxiety or depression in clinical settings [26], researchers have recently developed transdiagnostic treatments to address the underlying factors associated with both depression and anxiety [39,40]. However, it has been suggested that treatments developed for depression or anxiety alone may also have cross-over effects on comorbid symptoms. In fact, clinical research confirms that many treatments developed for depression also result in decreased symptoms of anxiety. One meta-analysis [41] reviewing randomized clinical treatment or prevention trials found that crossover effects were present in both depression treatments and anxiety treatments, although the effects were stronger for the targeted symptoms. That is, depression treatments had a greater effect on depressive symptoms than anxiety symptoms, and anxiety treatments had a greater effect on anxiety symptoms than depressive symptoms. However, there were no significant cross-over effects of depression or anxiety prevention interventions.

To address the need for preventive interventions that may reduce a range of symptoms in adolescents, CATCH-IT (Competent Adulthood Transition with Cognitive-behavioral, Humanistic, and Interpersonal Training), an online prevention intervention, was developed as a low cost, universally available intervention. Disseminated through primary care settings, CATCH-IT uses a population-health based model (screening and outreach in settings where youth and their families usually receive care) [42]. CATCH-IT contains elements of CBT and interpersonal psychotherapy in a highly interactive interface, which is described in more detail in previous publications [42,43]. Early pilot studies demonstrated that participants who used CATCH-IT reported decreases in symptoms of depression at 6, 12, and 30 months, relative to baseline [13,44,45]. A more recent randomized clinical trial found CATCH-IT to be efficacious in preventing depressive episodes at six months, relative to an attention-control condition that combined usual care with monitoring and an online general health education program (HE) in adolescents who were experiencing elevated symptoms of depression, though not in the full intention to treat analysis [12]. These initial findings suggest that CATCH-IT may have great potential as a public health intervention for adolescents at risk of developing Major Depressive Disorder (MDD). As an online intervention, CATCH-IT is easy to access, cost-effective, decreases stigmatization through anonymity, and does not require highly trained professionals [46].

Even though no cross-over effects on anxiety were reported for depression prevention interventions, there is some indication that CATCH-IT may affect symptoms of anxiety as well as depression. A study of CATCH-IT adapted for Chinese adolescents found that participants in the CATCH-IT condition demonstrated a decrease in symptoms of both depression and anxiety at eight-months [47]. While CATCH-IT is specifically designed for depression prevention, many of the skills it teaches may affect transdiagnostic factors, and therefore, decrease symptoms of anxiety as well. For example, cognitive behaviorally-based A-B-C exercises presented in CATCH-IT may increase emotional awareness, and the cognitive restructuring taught in CATCH-IT may decrease negative cognitive biases as well as negative affectivity. These effects could positively impact symptoms of anxiety in addition to depression. Given the accessibility, reach, and potential public health impact of the CATCH-IT intervention, it is important to investigate and understand the potential cross-over effects that CATCH-IT may offer.

The current study uses data from the randomized clinical trial comparing CATCH-IT to HE, an online program providing general health education, in a sample that was composed of adolescents with an intermediate to high risk of depression recruited from primary care settings [43]. As mentioned above, at six months, CATCH-IT, compared to HE, prevented depressive episodes in high-risk adolescents with significant subthreshold depressive symptoms at baseline [12], and additionally, depressive symptoms significantly decreased in both conditions. Moreover, preventive effects for CATCH-IT were found for up to 24 months for those with low levels of hopelessness or adequate levels of paternal monitoring [48]. The current study expands these initial findings to examine the possibility that the CATCH-IT intervention has long-term preventive effects on symptoms of anxiety as well as on symptoms of depression, and also to examine moderators of intervention effects on symptoms of both disorders. We hypothesize that participants in the CATCH-IT condition will have experienced greater decreases in self-reported symptoms of anxiety and depression, relative to participants in the HE group. Secondary analyses will examine potential moderators that may affect findings across both groups.

## 2. Materials and Methods

### 2.1. Study Design and Setting

This study utilized a hybrid type I efficacy-implementation design to evaluate the effectiveness of CATCH-IT in a scalable setting [49]. The efficacy of CATCH-IT was evaluated while simultaneously gathering information on CATCH-IT’s implementation to prepare for scaling up [12]. This trial was implemented in eight major US Health Systems (*N* = 31 primary care sites, 42,310 adolescents) in a population health approach (screen all youth, offer intervention, assessment, refer those in need of treatment) with over 1200 primary care staff consented and trained. This process has been previously described in further detail [43,50]. Data were collected from urban and suburban clinics located in Chicago, Illinois, and surrounding areas including northern Indiana, and Boston, Massachusetts and surrounding areas. The protocol, CONSORT statement, and a description of the implementation process have been described previously [43,50]. Institutional review board (IRB) approval was received from the University of Chicago IRB (protocol # 2011-0505), the IRB of record, on 15 December 2011, and also by local review boards. A data safety and monitoring board (DSMB) reviewed the trial and results twice per year.

### 2.2. Participants

The final sample included 369 adolescents, ages 13 to 18 years old, who were at risk for depression as indicated by elevated symptoms of depression (score of 8–17 on the 10-item or score of ≥16 on the 20-item Center for Epidemiologic Studies Depression scale (CES-D)) and/or a history of a major depressive episode or dysthymia. Exclusion criteria included a DSM-IV diagnosis of major depressive disorder (MDD), bipolar disorder, schizophrenia, psychosis, or current drug/alcohol abuse. Additionally, adolescents were excluded if they were in current treatment for MDD, had a significant developmental disability or reading impairment, were assessed to be at imminent risk of suicide, or were being treated for serious medical conditions [13,44]. 

### 2.3. Study Procedures

Participants were recruited from February 2012 to July 2016 through posted flyers, recruitment letters sent by primary care offices (sent to all families of adolescents in the targeted age range), and information given directly by the primary care physician (PCP) during clinic visits. Following verbal consent from a parent, the study staff administered a phone screen to assess adolescents for eligibility (Figure 1). Eligible adolescents and their parents attended an enrollment appointment at the office of their PCP. Informed consent was obtained before the completion of the baseline assessment. Depression risk was assessed by the KSADS, a clinician-administered, semi-structured interview. All baseline assessments were completed before informing the participants, their parents, or their PCPs of the intervention assignment. Follow-up assessments were conducted at 2, 6, 12, 18, and 24 months following baseline.

### 2.4. Randomization and Blinding

A computer-generated sequence blocked by site and time of entry was used to randomly assign participants to CATCH-IT or HE. Participants were also stratified by age (13–14 years vs. 15–18 years), gender, and severity of their risk of a depressive episode (high risk: elevated score on the Center for Epidemiological Studies-Depression Scale (CES-D_10_) and prior depressive episode; low risk: elevated CESD or prior depressive episode). The assignment of participants to HE versus CATCH-IT was 1:1. Assessors were blinded for the duration of the study, while investigators were blinded until all 12-month data were collected. PCPs were not blinded, since they provided motivational interviews to the adolescent participants in the CATCH-IT intervention.

### 2.5. Treatment Arms

#### 2.5.1. The CATCH-IT Intervention

CATCH-IT consists of 14 online modules designed to help adolescents develop coping strategies to decrease symptoms related to depression and prevent depressive episodes, plus an additional, optional module to address symptoms of anxiety. CATCH-IT was adapted from the Coping with Depression Adolescent Course, a group cognitive-behavioral intervention. It also includes elements of behavioral activation strategies as well as Interpersonal Psychotherapy techniques [51,52,53]. The program was designed using the Instructional Design Theory to maximize learning and maintain the learner’s attention [53]. In addition to the online modules, phone coaching by research staff (one to three calls) and a series of three motivational interviews conducted by the PCP (baseline, 2 months, and 12 months) were provided to support participants’ use of CATCH-IT. Parents were also invited to complete four online modules to support their child’s skills development, as well as an optional module for parents who worry they may themselves be depressed. Parent modules have been described in detail in previous publications [12,43,54].

#### 2.5.2. The Health Education Intervention

The HE intervention consists of 14 online modules providing general health information, and it was used as the control intervention. The first 13 modules of the HE program provide general health information, while the final module focuses on identifying mood symptoms, the importance of treating mental health issues, and how mental health stigma may interfere with seeking treatment. In addition to the online modules, up to three check-in calls were provided to each HE participant to ensure that they had access to the website within three weeks of the study enrollment. Parents were also invited to complete four modules consisting of similar information to the adolescent modules.

### 2.6. Measures

Measures were chosen to capture the range of ways that internalizing symptoms are expressed by adolescents and possible mediators and moderators of intervention response. All measures were descriptive and were appropriate for adolescents ages 13–18.

Demographics. Demographic information including sex, race, ethnicity, and maternal education was collected from adolescents and their parents at baseline.

Kiddie Schedule for Affective Disorders and Schizophrenia (KSADS). The KSADS is a semi-structured, clinician-administered, clinical interview that was used to determine if participants met the criteria for depression and/or were experiencing suicidal thinking [55]. Additionally, the KSADS was used to assess global functioning (GAS score) at baseline. The GAS score can range from 1–100, with higher scores indicating higher functioning.

Center for Epidemiological Studies-Depression Scale (CES-D_10_). The CES-D_10_ is a 10-item self-report or clinician-administered measure that assesses symptoms of depression over the past week. Scores can range from 0–30, with higher scores indicating higher levels of depressive symptoms. Adolescents and parents completed the CES-D_10_ at all time points [56]. Internal consistency of the CES-D in this sample was α = 0.71 for youth report, and α = 0.84 for parent report.

Screen for Child Anxiety Related Emotional Disorders (SCARED). The SCARED is a 41-item self-report measure that assesses symptoms of anxiety. Scores can range from 0–82, with higher scores indicating higher levels of anxiety symptoms [57]. The internal consistency of the SCARED in this sample was α = 0.91 for youth report.

Disruptive Behaviors Disorder Scale (DBD). The DBD is a 41-item self-report measure that assesses behavioral concerns in adolescents. The Attention Deficit Disorder (ADHD) subscale, which includes both the Inattentive and Hyperactivity/Impulsivity domains, and the Oppositional Defiant Disorder/Conduct Disorder (ODD/CD) subscale of the DBD can range from 0–3, with a higher score indicating more externalizing behavior [58]. Internal consistency for youth report in this sample was α = 0.84 for the ADHD subscale, and α = 0.87 for the ODD/CD subscale.

Social Adjustment Scale (SAS-SR). The SAS-SR is a 36-item self-report measure that assesses social functioning over the past two weeks. Scores can range from 0–4 with a higher score indicating a lower level of functioning [59]. The internal consistency of the SAS-SR in this sample was α = 0.79 for youth report.

Beck Hopelessness Scale (BHS). The BHS is a 21-item self-report measure that assesses pessimism about the future. Scores can range from 0–20, with a higher score indicating greater hopelessness [60]. The internal consistency of the BHS in this sample was α = 0.82 for youth report.

Child Report of Parental Behavior Inventory (CRPBI). The CRPBI is a 15-item measure that assesses a child’s relationship with their parents. Six subscales were used in this study: maternal and paternal acceptance (range = 10–30), psychological control (range = 8–24), and monitoring (range = 5–15). Higher scores indicate higher acceptance, control, and monitoring [61]. Internal consistency for youth report in this sample was α = 0.91 for maternal acceptance, α = 0.78 for maternal control, α = 0.80 for maternal monitoring, α = 0.92 for paternal acceptance, α = 0.82 for paternal control, and α = 0.87 for paternal monitoring.

Theory of Planned Behavior (TPB). The TPB Scale is a 19-item self-report measure that assesses attitude toward participating in a depression prevention intervention. Scores range from 1–5, with a higher score indicating a more positive attitude [62]. The internal consistency of the TPB in this sample was α = 0.87 for youth report.

Trans-Theoretical Model. The Trans-Theoretical Model Scale is a 10-item self-report measure that asks the adolescent to rate the importance of preventing an episode of clinical depression, plus their ability and readiness to reduce their depression risk. Scores can range from 1–10, with a higher score indicating a higher overall intention to prevent future episodes [63]. The internal consistency of the Trans-Theoretical Model Scale in this sample was α = 0.86 for youth report.

The Adolescent Life Events Questionnaire (ALEQ). The ALEQ is a retrospective self-report measure that asks about life events in the past 6 months. Scores can range from 0–51, with a higher score indicating more life events [64].

Engagement. Engagement in each of the interventions was assessed by the number of modules completed by parents and the number of modules completed by adolescents. A third variable was created to assess overall family engagement by summing the number of modules completed by an adolescent with the number of modules completed by the parent.

Positive Relationships in Primary Care (PRPC). The PRPC was administered at two months and asked the adolescent about their most recent interview with their primary care provider. For CATCH-IT participants, this may have been a motivational interview; for HE participants, it was their last PCP visit. The measure included both general questions (“I feel the primary care provider listened to me”) and questions specific to the intervention (“It was helpful to focus on behaviors I would like to change.”) Scores can range from 1–5, with higher scores indicating a more positive rating [44,65].

### 2.7. Statistical Analysis

Linear mixed-effect growth models with random intercept and slope were used to examine differences between groups in change over time in CES-D_10_ and SCARED scores. The CES-D_10_ analysis was conducted with and without a square root transformation of the time scale to improve linearity. All models were adjusted for sex, ethnicity (Hispanic, non-Hispanic), race (white, non-white), baseline age, and site, and the SCARED analysis was also adjusted for baseline teen CES-D_10_. The *p*-value for the group*time interaction was used to test for a significant difference in slopes between CATCH-IT and HE. Within-group changes over time were estimated using simple slopes, and adjusted mean change over 24 months was calculated by multiplying the estimated slope by 24 (or the square root of 24 in the time-transformed model).

We also examined the moderating effects of theory-based covariates by including interaction terms in the models. The *p*-value of the group * time * moderator interaction was used to test for potential moderation of the effect of the intervention. For both the CES-D_10_ and SCARED outcomes, we examined the following potential moderators: demographics (sex, race, ethnicity, maternal education), site (Chicago or Boston), engagement (modules completed by teens, parents, teens + parents), PRPC scores at two months, and baseline scores for GAS, CRAFFT, DBD (ADHD, ODCD), SAS-SR, Theory of Planned Behavior, ALEQ, BHS, Trans-Theoretical Model, Parent CES-D_10_, and the CRPBI (3 maternal and 3 paternal scales). We also tested SCARED as a potential moderator of the CES-D_10_ outcome and vice versa. In the moderator models with CES-D_10_ as the outcome variable, time was square-root transformed to improve linearity.

Completion or use of the supplementary CATCH-IT anxiety module was not included as a potential moderator in the analyses since only 18 (9.3%) teens completed the module, simply not a large enough percentage of the sample to allow us to test the effects of the module on anxiety and depressive symptoms. Additionally, our models included both CATCH-IT and HE participants to allow us to look for between-group differences. Therefore, we were unable to include variables that only applied to CATCH-IT as either covariates in the model or potential moderators.

Multivariable logistic regression models were used to test for differences between participants with and without CES-D_10_ or SCARED data at 12 and 24 months, including the following covariates: site, intervention group, age at baseline, gender, ethnicity, race, maternal education, parents’ marital status, birth order (firstborn, other), past depressive episode at study entry, and high CES-D score at study entry. All analyses were conducted using SAS, version 9.4 (SAS Institute, Cary, NC, USA).

## 3. Results

### 3.1. Participants

One hundred ninety-three participants were randomized to CATCH-IT and 176 to HE. The mean age of the participants was 15.4 years (SD = 1.5), and 21% of the participants identified as Hispanic, 26% non-Hispanic black, 43% non-Hispanic white, 4% Asian, and 6% multiracial or other. Over half of the adolescents’ parents reported having no more eduction than a high school diploma (60% of mothers and 53% of fathers), and 39.4% reported not being married. Clinically, 62% of participants reported a past sub-threshold or full depressive episode that had ended at least two months before study enrollment. Participants reported a mean score of 25.3 (SD = 12.3) on the SCARED, and a mean score of 9.4 (SD = 4.6) on the CES-D_10_ at baseline.

Site differences were found between the Chicago and Boston participants. Participants from Boston reported higher levels of parent education and a greater likelihood that parents were married. Additionally, the Boston sample was comprised of a lower percentage of ethnic minority participants, and a greater percentage of adolescents in Boston qualified for the study based on only a prior depressive episode. A complete description of the cohort data at baseline is provided in a prior publication [12].

Regarding attrition over the study period, at 24 months CESD scores were available for 182 participants (49.3%), and SCARED scores were available for 93 participants (25.2%). For the CES-D_10_, missing scores at 24 months were associated with CATCH-IT assignment, the Chicago site, higher age at baseline, lower parent education, and elevated depressed mood at baseline (see Appendix A); for the SCARED, missing scores at 24 months were associated with CATCH-IT assignment, the Chicago site, and lower maternal education (see Appendix A).

### 3.2. Symptoms of Anxiety and Depression

There was not a significant difference between groups in change over time for either the SCARED (*p* = 0.79) or the CES-D_10_ (*p* = 0.80), see Table 1. Anxiety appeared to improve in both groups, though the change was only statistically significant in CATCH-IT. The estimated slope for SCARED scores was −0.13, SE = 0.06, *p* = 0.04 for CATCH-IT and −0.11, SE = 0.06, *p* = 0.07 for HE. The estimated mean change from baseline to 24 months was −3.1 (SE = 1.5) points in the CATCH-IT group and −2.6 (1.4) points in the HE group. Depressive symptoms improved significantly in both groups. The estimated slope for the CES-D_10_ scores with time square-root transformed was −0.31, *p* = 0.004 for CATCH-IT and −0.27, *p* = 0.009 for HE. The estimated mean change from baseline to 24 months was −1.5 (SE = 0.5) points for CATCH-IT and −1.3 (0.5) points for HE.

### 3.3. Analyses of Potential Moderators

Symptoms of Anxiety. Two significant moderators of the effect of CATCH-IT and HE on change in SCARED scores across the 24-month study period were identified: parental CES-D10 scores at baseline (b = −0.05, *p* = 0.004) and positive relationships in primary care scores at 2 months (b = −0.41, *p* = 0.009) (Table 2). Anxiety appeared to decrease more in the HE group than in the CATCH-IT group when parental depressive symptoms at baseline were very low. When parental CES-D10 was set to 0 in the model, the estimated slope for HE was −0.25 (SE = 0.09), and the estimated slope for CATCH-IT was 0.04 (0.09), *p* = 0.03 (see Appendix A). However, anxiety decreased more in the CATCH-IT group than the HE group when parental depressive symptoms at baseline were very high. When parental CES-D10 was set to 27, the maximum score in our sample, the estimated CATCH-IT slope was −0.62 (0.23), while the estimated HE slope was 0.36 (0.26), *p* = 0.005. At intermediate parental CES-D10 scores (25th percentile, median, 75th percentile), the model did not show significant differences between groups in anxiety change.

Similarly, anxiety appeared to show greater improvement in the HE group than the CATCH-IT group when adolescents gave a very low score to their most recent interaction with their primary care provider (see Appendix A). When PRPC was set to 1.0 in the model, the estimated slope for HE was −1.05 (SE = 0.31), while the estimated slope for CATCH-IT was 0.20 (0.37), *p* = 0.01. However, when PRPC was set to the maximum score of 5.0, the estimated slope was 0.18 (0.14) for HE but −0.22 (0.14) for CATCH-IT, *p* = 0.050. The model did not predict significant differences in anxiety change between groups at intermediate PRPC scores.

Symptoms of Depression. Three significant moderators of the effect of CATCH-IT and HE on change in adolescent CES-D10 scores across the 24-month study period were identified: baseline DBD-ADHD subscale score (b = 1.42, *p* = 0.004), baseline DBD-ODD/CD subscale score (b = 2.37, *p* = 0.01), and PRPC score at two months (b = −0.60, *p* = 0.046) (Table 3). In the ADHD moderator model, CATCH-IT appeared superior to HE in reducing CES-D10 scores when ADHD scores were low, but HE appeared superior when ADHD scores were high (see Appendix A). When the ADHD score was set to 0 in the model, the estimated slope for CATCH-IT was −0.31 (SE = 0.28), while the estimated slope for HE was 0.89 (0.32), *p* = 0.006. CATCH-IT still appeared superior when ADHD was set to the approximate 25th percentile in our sample (0.5), though the difference between slopes was smaller: −0.25 (0.16) for CATCH-IT vs 0.24 (0.18) for HE, *p* = 0.04. However, when ADHD was set to the maximum value for our sample (2.0), the estimated CES-D10 slope was only −0.06 (0.43) for CATCH-IT but −1.70 (0.44) for HE, *p* = 0.008. For ADHD scores at the median and 75th percentiles for our sample, the model did not predict a significant difference between groups.

Results were similar for the DBD-ODCD subscale (see Appendix A). When ODCD was set to 0 in the model, the estimated improvement in depressive symptoms was greater in the CATCH-IT group: slope = −0.29 (0.19) for CATCH-IT and 0.38 (0.23) for HE, *p* = 0.03. When ODCD was set to the maximum score in our sample (still quite low at 1.25), the estimated slopes were 0.18 (0.71) for CATCH-IT and −2.11 (0.65) for HE, *p* = 0.02. The model did not predict significant differences between slopes at intermediate values of ODCD.

As was noted for anxiety, the HE group showed a greater estimated reduction in depressive symptoms than the CATCH-IT group when the PRPC rating was very low (see Appendix A). With PRPC set to 1 (minimum score) in the model, the estimated CATCH-IT slope was 0.76 (0.70), but the HE slope was −1.22 (0.58), *p* = 0.03. However, the model did not show significant differences between groups at any other PRPC value tested, including the maximum score of 5.

## 4. Discussion

In a long-term follow-up investigation, the current study examined symptoms of anxiety and depression in at-risk adolescents who were recruited through primary care and were assigned randomly to the CATCH-IT prevention intervention or HE. Consistent with data from a 6-month follow-up of this sample [12], adolescents assigned to both the CATCH-IT intervention and the HE intervention experienced decreasing depressive symptoms over the course of 24 months. Moreover, differences in score trajectories were not found between the conditions, suggesting that both interventions had a positive effect on depressive symptoms. The multiple study contacts (i.e., screenings, assessments, safety calls) that accompanied both intervention approaches may have obscured group differences. In addition, while the content of the HE program focused on health behaviors and not on mental health, participants and parents may have received some sense of mastery and self-efficacy from completing the program, which may account for lower depression scores for HE participants.

The current study also provided evidence for cross-over effects in the CATCH-IT group. Although there was not an overall group difference in anxiety symptoms, teens at risk for depression who were assigned to CATCH-IT experienced a significant decrease in symptoms of anxiety across the 24 months, while there was no significant change in anxiety symptoms for teens assigned to HE. This finding is consistent with results from Ip et al. [47], who found that adolescents who used a Chinese adaptation of the CATCH-IT prevention intervention, Grasp the Opportunity, reported fewer symptoms of anxiety at 8 months, relative to adolescents assigned to a control (attention) intervention. Evidence of cross-over effects from CATCH-IT suggests that this prevention intervention, which was developed to address risk for depression only, may in fact target shared underlying mechanisms for both depression and anxiety. These results are counter to Garber et al. [41], who, in a meta-analysis exploring cross-over effects for depression and anxiety treatment and prevention interventions, reported no evidence of cross-over effects for depression prevention programs. It is possible that the recruitment process, intervention engagement, and motivational interviews that accompany the CATCH-IT intervention, all of which emphasize building resilience, managing stressors, and working toward goals in addition to preventing low mood and depression, may help teens to generalize depression prevention skills to other distressing symptoms such as anxiety. It is also true that the CATCH-IT intervention includes an optional anxiety module (Module #15), which addresses symptoms of anxiety more directly and provides instructions for exercises such as progressive muscle relaxation. Because so few adolescents (*N* = 18) in our sample completed the anxiety module, however, it is more likely that the observed improvement in anxiety symptoms is explained by participants’ use of other skills taught in CATCH-IT, such as cognitive restructuring and behavioral activation, both of which are commonly used to treat symptoms of anxiety in adolescents [66,67,68].

Secondary analyses revealed that parental depression moderates the trajectory of anxiety symptoms in teens assigned to CATCH-IT compared to HE. Specifically, when parents had no or low-level symptoms of depression at baseline, their adolescents reported a greater improvement in anxiety when assigned to HE; when parents had high-level symptoms of depression at baseline, their adolescents reported a greater improvement in anxiety when assigned to CATCH-IT. This finding, that CATCH-IT is associated with better outcomes for teens when parents report high-level depressive symptoms, contrasts with prior research suggesting that parental depressive symptoms are associated with *poorer* intervention outcomes for children with internalizing symptoms [69,70]. Our findings are consistent, however, with research reporting better youth intervention outcomes when parents report symptoms of anxiety [71,72,73], and may reflect the additional motivation symptomatic parents feel to support their children’s engagement in the CATCH-IT intervention. It is also possible that parents with depressive symptoms were more engaged in the CATCH-IT parent intervention, and that their use of the on-line parent modules, as well as the motivational interviews focusing on their goals for supporting their child in completing CATCH-IT, improved their child’s long-term outcomes. In fact, Legerstee and colleagues [71] suggested that anxious mothers may have benefitted from the parent intervention that accompanied their adolescent’s intervention, and therefore were able to support their teen’s engagement in treatment, leading to better outcomes over time. Unfortunately, baseline parental anxiety was not measured in this trial, but adolescents with elevated symptoms of depression at baseline were found to benefit more from CATCH-IT than non-symptomatic adolescents [12].

Additionally, the adolescent’s relationship with their primary care physician moderated the trajectory of anxiety symptoms in adolescents assigned to CATCH-IT relative to HE. Adolescents who reported a weak connection to their primary care physician experienced a greater improvement in anxiety when they were assigned to HE rather than CATCH-IT. The benefits adolescents received from the HE intervention are likely a function of the non-specific factors associated with study involvement (e.g., recruitment, assessments, check-in phone calls, sense of achievement from completing modules), and perhaps these non-specific factors were more motivating for adolescents in the absence of a close connection with their primary care physician.

Secondary analyses also revealed some interesting moderating effects of adolescents’ externalizing behaviors on their depressive symptoms across the 24-month follow-up interval, such that, when adolescents reported more symptoms of attentional and oppositional behaviors, they experienced a greater improvement in depressive symptoms if they were assigned to the HE group. By extension, this finding suggests that the CATCH-IT intervention, relative to HE, may be less helpful to adolescents who are struggling with externalizing concerns, perhaps due to its high reading demand, and its demand for attention, homework completion, and compliance. Solanto [74] suggests that the self-management challenges for adolescents with ADHD may interfere with their response to cognitive-behavioral and self-directed interventions. Nevertheless, given the overall low mean score for this sample on the DBD, the measure of externalizing behaviors used in this trial, these moderating effects must be interpreted with caution. Similar to our findings on symptoms of anxiety, depressive symptoms decreased more across the 24 months for adolescents in HE than in CATCH-IT when they reported a weak connection to their primary care physician.

There are several limitations associated with this study. We experienced attrition in the sample over the 24-month study period, particularly among adolescents with fewer resources (e.g., Chicago site, higher depressed mood, lower parental education). It is noteworthy that, to our knowledge, this is the only Internet-based intervention trial for youth with a 2-year follow-up, and such attrition is to be expected when such a long interval follows an intervention with relatively minimal face-to-face interaction. It is also noteworthy that attrition was higher for adolescents assigned to CATCH-IT versus HE, despite the presence of physician contact through motivational interviews in the CATCH-IT condition only. It is difficult to account for this finding, but it is possible that adolescents who benefitted most from the CATCH-IT intervention were less likely to feel a need to remain connected with the study throughout the follow-up interval. To address attrition in our sample, we chose to use mixed effect models, which are thought to be robust to missing data. The use of a second, potentially active internet-based intervention as a control is another limitation to the study. While the content of HE was not focused on mental health (only a single module towards the end of the intervention addressed mental health concerns), the program did present information on other health behaviors. It is possible that teens assigned to the HE condition received benefit above what they would have in an inactive or waitlist control. In addition, the examination of very long-term intervention effects (24 months) is necessarily constrained by the possibility of intervening environmental factors (e.g., natural disasters, family disruption) that may limit the ability to draw conclusions regarding causality.

## 5. Conclusions

Despite these limitations, the current study demonstrates long-term reductions in symptoms of depression in both intervention groups (HE and CATCH-IT). Additionally, participants assigned to CATCH-IT were found to have cross-over effects as seen in a reduction of anxiety symptoms, demonstrating that it is possible to develop preventive interventions for adolescent depression that target transdiagnostic factors representing shared underlying mechanisms for both disorders. Similarly, moderation results of depression and anxiety symptoms reported here, in addition to earlier moderation results of depressive disorder outcomes [48], suggest that CATCH-IT is beneficial when certain conditions are present (i.e., no significant externalizing symptoms (depressed mood), no significant hopelessness (depressive episodes), a supportive relationship with a physician (anxiety and depressive symptoms), heightened parental symptoms (anxiety), and adequate paternal monitoring (depressive episodes)). That is, pre-conditions (current depressed mood), lack of substantial co-morbidity, and supportive environmental factors may be necessary for adolescents to benefit from this largely self-directed model. Given that comorbid presentations of depression and anxiety are common and are associated with poor long-term outcomes, the discovery of a single preventive intervention that targets both disorders has significant implications for youth mental health promotion. These findings, along with the ability to implement the program across multiple primary care clinics, demonstrates the value of CATCH-IT as a feasible, online, population-based depression prevention intervention for at-risk adolescents.

## Figures and Tables

**Figure 1 ijerph-17-07736-f001:**
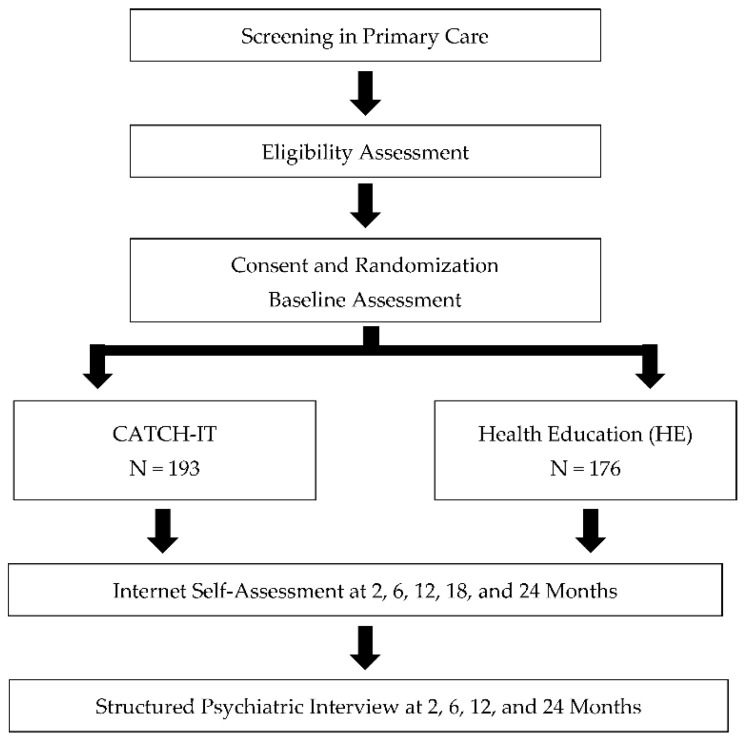
Study design.

**Table 1 ijerph-17-07736-t001:** Internalizing symptoms, baseline to 24 months.

	Unadjusted Means							Within-Group Slopes ^1^						Btw Grp Diff
Outcome Variables	CATCH-IT (*N* = 193)			HE (*N* = 176)				CATCH-IT			HE			
	N	Mean	SD	N	Mean	SD	Time Variable	b	SE	*p*	b	SE	*p*	*p*
SCARED ^2^ total score (0–82)							Months	−0.13	(0.06)	0.04	−0.11	(0.06)	0.07	0.79
Baseline	171	25.5	(12.7)	141	25.2	(11.9)								
2 months	94	26.2	(13.1)	90	25.7	(14.3)								
6 months	74	25.1	(15.0)	92	24.0	(13.9)								
12 months	83	22.4	(13.0)	73	23.2	(13.6)								
18 months	64	25.6	(13.3)	81	23.2	(13.4)								
24 months	39	19.6	(12.2)	54	20.9	(13.7)								
CES-D_10_ (0–30)							Months	−0.06	(0.02)	0.01	−0.04	(0.02)	0.10	0.44
Baseline	190	9.5	(4.5)	172	9.4	(4.6)	Sqrt (months)	−0.31	(0.11)	0.004	−0.27	(0.10)	0.009	0.80
2 months	123	9.2	(4.2)	140	7.9	(5.2)								
6 months	116	8.2	(4.9)	133	7.9	(4.7)								
12 months	115	7.9	(5.3)	126	7.1	(4.7)								
18 months	67	7.9	(5.9)	81	7.5	(5.6)								
24 months	79	7.7	(5.8)	103	8.1	(5.6)								

^1^ From linear mixed effect growth models with random intercept and slope, adjusted for sex, ethnicity (Hispanic, non-Hispanic), race (white, non-white), baseline age, site, and baseline teen CES-D_10_. Within-group estimated slopes and *p*-values are from estimates of simple slopes. The *p*-value for the visit*time interaction is used to test for a significant difference between slopes. ^2^ Higher scores indicate greater anxiety.

**Table 2 ijerph-17-07736-t002:** Summary of moderator analyses for Screen for Child Anxiety Related Emotional Disorders (SCARED): Group*Visit*Moderator Interaction Term.

Moderator	*N* ^1^	Beta	Standard Error	*p* ^2^
Sex = male (reference = female)	362	0.16	0.19	0.41
Race = not white (reference = white) ^3^	362	0.14	0.18	0.43
Ethnicity = Hispanic (reference = not Hispanic)	362	−0.17	0.25	0.50
Site = Boston (reference = Chicago)	362	0.29	0.17	0.09
Maternal education= college degree (reference = no college degree)	352	0.15	0.19	0.43
Teen GAS, baseline	360	0.00	0.01	0.66
Teen CES-D_10_, baseline	362	0.02	0.02	0.17
Parent CES-D_10_, baseline	332	−0.05	0.02	0.004
ADHD (DBD-A), baseline	196	0.12	0.26	0.64
ODCD (DBD-A), baseline	193	0.28	0.52	0.59
Social adjustment (SAS-SR), baseline	212	0.15	0.23	0.51
Hopelessness (BHS), baseline	270	−0.01	0.03	0.74
Maternal acceptance (CRPBI), baseline	196	−0.04	0.02	0.06
Maternal control (CRPBI), baseline	197	0.04	0.03	0.18
Maternal monitoring (CRPBI), baseline	187	−0.10	0.06	0.12
Paternal acceptance (CRPBI), baseline	176	−0.04	0.02	0.07
Paternal control (CRPBI), baseline	176	0.00	0.03	0.88
Paternal monitoring (CRPBI), baseline	170	−0.04	0.04	0.25
Positive relationships in primary care, 2 months	133	−0.41	0.16	0.009
Theory of planned behavior, baseline	164	−0.47	0.25	0.06
Stressful life events (LEQ), baseline	300	−0.02	0.02	0.24
Trans-theoretical model, baseline	191	−0.07	0.05	0.12
Teen modules completed	362	0.00	0.01	0.83
Parent modules completed	340	−0.03	0.04	0.45

^1^ Number of participants included in the analysis. Participants with missing data for the moderator variable or baseline CESD (covariate) were excluded. ^2^ From linear mixed effect growth models with random intercept and slope and a group*visit*moderator interaction term, adjusted for sex, ethnicity (Hispanic, non-Hispanic), race (white, non-white), baseline age, site, and baseline teen CES-D_10_. ^3^ No significant effects of race were detected when the race categorization was changed to Black or multi-racial vs. all others.

**Table 3 ijerph-17-07736-t003:** Summary of moderator analyses for CESD: Group*Visit*Moderator Interaction Term.

Moderator	*N* ^1^	Estimate	SE	*p* ^2^
Sex = male (reference = female)	369	0.06	0.33	0.86
Race = not white (reference = white)	369	0.54	0.30	0.07
Ethnicity = Hispanic (reference = not Hispanic)	369	−0.09	0.38	0.81
Site = Boston (reference = Chicago)	369	0.01	0.31	0.98
Maternal education = college degree (reference = no college degree)	359	0.03	0.31	0.91
Teen GAS, baseline	367	0.00	0.02	0.93
Anxiety (SCARED), baseline	312	0.00	0.01	0.99
Parent CES-D_10_, baseline	338	0.01	0.03	0.84
ADHD (DBD-A), baseline	196	1.42	0.48	0.004
ODCD (DBD-A), baseline	193	2.37	0.93	0.01
Social adjustment (SAS-SR), baseline	212	−0.22	0.45	0.64
Hopelessness (BHS), baseline	270	0.03	0.05	0.55
Maternal acceptance (CRPBI), baseline	196	−0.04	0.05	0.40
Maternal control (CRPBI), baseline	197	0.07	0.06	0.23
Maternal monitoring (CRPBI), baseline	187	0.02	0.10	0.86
Paternal acceptance (CRPBI), baseline	176	−0.05	0.04	0.21
Paternal control (CRPBI), baseline	176	0.06	0.06	0.35
Paternal monitoring (CRPBI), baseline	170	−0.07	0.07	0.29
Positive relationships in primary care, 2 months	134	−0.60	0.30	0.046
Theory of planned behavior, baseline	164	−0.91	0.50	0.07
Stressful life events (LEQ), baseline	302	0.05	0.03	0.10
Trans-theoretical model, baseline	191	0.08	0.10	0.42
Teen modules completed	369	0.00	0.03	0.88
Parent modules completed	346	−0.05	0.08	0.52

^1^ Number of participants included in the analysis. Participants with missing data for the moderator variable were excluded. ^2^ From linear mixed effect growth models with random intercept and slope and a group*visit*moderator interaction term, adjusted for sex, ethnicity (Hispanic, non-Hispanic), race (white, non-white), baseline age, and site. Time square-root transformed to improve linearity.

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
