# Peer review of "Randomized Clinical Trial of an Internet-Based Adolescent Depression Prevention Intervention in Primary Care: Internalizing Symptom Outcomes"

_ijerph, 2020, doi:10.3390/ijerph17217736_

Round 1

Reviewer 1 Report

Thank you for the opportunity to read this interesting paper. I found it clinically relevant. I have a few items for minor revisions that I wanted to raise with you.

The paragraph transition at lines 84/85 is abrupt moving us from the background literature review into the focus of the research. Some bridging seems needed here.

Line 113 - HE - this is the first time this is introduced. It is explained later Iline 180) but might be better to at least explain HE. 

Line 148 - the data is now 5 years old. It appears there have been several other papers from this research. What makes this paper unique and offers insights not offered in the prior publications.

There are many measures - why so many? There is likely a good explanation but it would help the reader to know that. It would also help the reader to know that each measure is age appropriate to the population and are these descriptive measures or normed to specific populations?

Line 240 - Can you explain why the focus upon Hispanic - non-Hispanic - again I suspect there is a likely explanation but that will help the reader. We see this in Table 2 for example but Hispanic represents 21% of the sample and non-Hispanic Black 26%. I was left wondering why the Black population was not separately considered. While I note at line 277/8 there is reference to the full cohort data in a prior publication, it feels like a gap in this paper.  I have read the prior paper and did not find an explanation for the Hispanic emphasis. 

Line 429 - In looking at the prior paper, attrition seems to have been reasonably significant. I wonder about a line here giving more detail about the loss rather than simply noting attrition. 

In terms of limitations, I also wonder if it might be wise to speak about one of the significant weaknesses that the longer out a study goes, the more challenge with confounding environmental factors. This tends to limit even causal conclusions. 

The paper is very much worthy of publication. Preparing multiple papers from a project is important. In my comments I have tried to also focus that many readers will only see this version of the project data so I am trying to ensure that the paper accomplishes the goal within this journal. 

Author Response

Point 1: The paragraph transition at lines 84/85 is abrupt moving us from the background literature review into the focus of the research. Some bridging seems needed here.

Response 1: We agree and have added some transitional language (line 86) to connect our review of the literature to the focus of the current manuscript.

Point 2: Line 113 - HE - this is the first time this is introduced. It is explained later Iline 180) but might be better to at least explain HE. 

Response 2: We have added a brief explanation of the HE intervention the first time it is introduced (line 115-116).

Point 3: Line 148 - the data is now 5 years old. It appears there have been several other papers from this research. What makes this paper unique and offers insights not offered in the prior publications.

Response 3: We have added a sentence to the last paragraph of the introduction (lines 122-125) explaining the unique contribution of this paper, which expands earlier reports of preventive effects on depressive episodes to focus on the exploration of cross-over effects for symptoms of anxiety and depression.

Point 4: There are many measures - why so many? There is likely a good explanation but it would help the reader to know that. It would also help the reader to know that each measure is age appropriate to the population and are these descriptive measures or normed to specific populations?

Response 4: We recognize that we included many measures in this study. We chose to do so because we wanted to capture the many ways that depression and depression risk are expressed by adolescents, and also so we could explore a range of mediators and moderators for any intervention effects we may discover. We also were interested in capturing adolescents’ response to the intervention itself, and any possible explanations for why adolescents may use the intervention more or less. We briefly addressed this just prior to introducing our list of measures (lines 199-201). In the same paragraph, we also noted that all measures were appropriate for the age of our sample, and also that the measures were descriptive in nature.

Point 5: Line 240 - Can you explain why the focus upon Hispanic - non-Hispanic - again I suspect there is a likely explanation but that will help the reader. We see this in Table 2 for example but Hispanic represents 21% of the sample and non-Hispanic Black 26%. I was left wondering why the Black population was not separately considered. While I note at line 277/8 there is reference to the full cohort data in a prior publication, it feels like a gap in this paper.  I have read the prior paper and did not find an explanation for the Hispanic emphasis. 

Response 5: Thank you for this thoughtful comment. To be consistent with our analyses in other publications, we coded the race variable that we used in our models as white/all others. However, to address this concern, we did examine the breakdown of our sample by race only (without including ethnicity), and we learned that our sample is 28.4% Black.  We further ran our primary moderation analyses for Black participants only (Black or multi-racial vs. all others), without ethnicity, and found that race remained a non-significant moderator of the relation between intervention group and change over time for either anxiety or depressive symptoms.  We have noted this in a footnote to Table 2 (line 349-350).

Point 6: Line 429 - In looking at the prior paper, attrition seems to have been reasonably significant. I wonder about a line here giving more detail about the loss rather than simply noting attrition. 

Response 6: We have added several sentences to our Discussion section (lines 467-476) to address the issue of attrition.  We explained that attrition was higher among participants with fewer resources, provided an explanation for the high rates of attrition we found, and also addressed the issue that higher rates of attrition were found for CATCH-IT versus HE participants. 

Point 7: In terms of limitations, I also wonder if it might be wise to speak about one of the significant weaknesses that the longer out a study goes, the more challenge with confounding environmental factors. This tends to limit even causal conclusions. 

Response 7: Thank you for this important contribution.  We have added a sentence to our Discussion section (line 482-484) that addresses the possibility that, over such a long follow-up interval, environmental factors may impact our outcomes.

Reviewer 2 Report

Title: Randomized Clinical Trial of an Internet-Based Adolescent Depression Prevention Intervention in Primary Care: Internalizing Symptom Outcomes

The study investigates the effects of CATCH-IT, an online depression prevention intervention, compared to online health education (HE) on internalizing symptoms in adolescents at risk for depression. The participants were recruited across different US health systems and randomly assigned to the two conditions. Assessments were completed at baseline, two, six, 12, 18, and 24 months. No significant differences between groups were found with respect to change in depressive and anxiety symptoms. Improvement in depressive symptoms was significant for both groups whereas improvements in anxiety was significant only for CATCH-IT Group. Findings were moderated by different variables (e.g. parental depression, relationships with primary care physicians and externalizing symptoms). The authors conclude suggesting the long-term positive effects of both online programs on depressive symptoms, and a cross-over effect on anxiety for the CATCH-IT. I have found the article of particular interest with respect to the possibility to implement targeted and evidence-based interventions in adolescence, especially in an online format that could enhance the subsequent scaling up phase. The manuscript is well written; the theoretical background supports the research hypotheses and sections inherent methods and statistical elaboration of data/results are clear to understand. Following you can find few suggestions for minor revisions:

Introduction

(Line 126) please provide reference for the conceptualization of “hybrid type I efficacy-implementation design”

(Line 127) substitute “efficacy” with effectiveness

Material and methods

(Line 161) please report into which categories the risk of depressive episodes was stratified

Paragraph 2.6. Measures – please provide reliability values for each of the instruments (e.g. Cronbach’s alpha)

Paragraph 2.7. Statistical analyses –  Given the focus both on depression and anxiety, and in light of the results of the study, I was wondering whether, besides the engagement variables, the authors did control for how many individuals completed also the CATCH-IT supplementary anxiety module. I suggest to better specify this aspect or, otherwise, to provide rational for why not doing so.

Results

(Lines 279-280) A verb is probably missing in the sentence. Please revise.

Discussion

I would suggest making some considerations on the higher attrition for the CATCH-IT Group, also given that in this group PCPs provided motivational interviews to the adolescent participants.

(Lines 386-387) I would spend some more words on the effect that the CATCH-IT additional anxiety module could have with respect to the findings of the study

Author Response

Point 1: (Line 126) please provide reference for the conceptualization of “hybrid type I efficacy-implementation design”

Response 1: We have added a reference for this conceptualization (line 134). The reference is to a paper by Curran and colleagues (2012).

Point 2: (Line 127) substitute “efficacy” with effectiveness

Response 2: Thank you for this correction. We have made this substitution (line 134).

Point 3: (Line 161) please report into which categories the risk of depressive episodes was stratified

Response 3: We have added a description of the categories into which the risk of depressive episodes was stratified (line 170-171).

Point 4: Paragraph 2.6. Measures – please provide reliability values for each of the instruments (e.g. Cronbach’s alpha)

Response 4: We have now included the Chronbach’s alpha for each measure for which internal consistency reliability values can be calculated (because the life events measure is a count rather than a scale, we could not calculate the reliability coefficient for this measure) (lines 212-246).

Point 5: Paragraph 2.7. Statistical analyses –  Given the focus both on depression and anxiety, and in light of the results of the study, I was wondering whether, besides the engagement variables, the authors did control for how many individuals completed also the CATCH-IT supplementary anxiety module. I suggest to better specify this aspect or, otherwise, to provide rational for why not doing so.

Response 5: Thank you for raising this important issue. In fact, only 18 teens completed this optional module, too small of a group to enable us to test its effects on symptoms of anxiety and depression. Moreover, because our models compared participants in each group (CATCH-IT vs. HE), we could only examine variables that were present in both groups, and there was no Module 15 in HE. We have added an explanation for why we did not control for the number of participants who completed the optional anxiety module to our section describing our statistical analyses (lines 281-286).

Point 6: (Lines 279-280) A verb is probably missing in the sentence. Please revise.

Response 6: We apologize for this error. We have revised the sentence (lines 308-310).

Point 7: I would suggest making some considerations on the higher attrition for the CATCH-IT Group, also given that in this group PCPs provided motivational interviews to the adolescent participants.

Response 7: We have added consideration for the higher attrition in the CATCH-IT group to our Discussion section (lines 471-476).  Specifically, we noted the possibility that the teens who responded best to CATCH-IT may not have completed the long-term follow-up assessment. 

Point 8: (Lines 386-387) I would spend some more words on the effect that the CATCH-IT additional anxiety module could have with respect to the findings of the study

Response 8: We agree that the possible effects of the anxiety module should be addressed more fully in our discussion, and we have done so in lines 420-424. We explain that, because so few teens completed the optional anxiety module, it is unlikely that the decrease we found in anxiety symptoms can be attributed to completion of that module, and we suggest other skills taught throughout CATCH-IT that also are used commonly to treat symptoms of anxiety in adolescents.  

Reviewer 3 Report

Dear Authors,

Thank you for the interesting, well-structured, and well-written paper.

I have only minor comments.
I suggest explaining the abbreviation CATCH-IT when it is used in the abstract for the first time.
The introduction reveals the problem of a high prevalence of depression and adolescents and a need of cost-effective programs.
The selection of the study participants and assignment to the groups is described in a detailed way. My suggestion is to provide the final sample size in section 2.2 and consider the need of the flow chart illustrating the recruitment of the study participants.
Statistical methods are adequate. The strength of the analyses is taking into account effect moderators. The results are clearly presented. Study limitations are provided.

Best regards.

Author Response

Point 1: I suggest explaining the abbreviation CATCH-IT when it is used in the abstract for the first time.

Response 1: We have written out the abbreviation for CATCH-IT in the abstract (lines 17-18).

Point 2: The selection of the study participants and assignment to the groups is described in a detailed way. My suggestion is to provide the final sample size in section 2.2 and consider the need of the flow chart illustrating the recruitment of the study participants.

Response 2: We have provided the final sample size in section 2.2, as suggested (line 146), and we have added a flow chart (Figure 1) illustrating the recruitment of the study participants.